# Immunotherapy in Adrenocortical Carcinoma: Predictors of Response, Efficacy, Safety, and Mechanisms of Resistance

**DOI:** 10.3390/biomedicines9030304

**Published:** 2021-03-16

**Authors:** Marta Araujo-Castro, Eider Pascual-Corrales, Javier Molina-Cerrillo, Teresa Alonso-Gordoa

**Affiliations:** 1Neuroendocrinology Unit, Endocrinology and Nutrition Department, Ramón y Cajal Health Research Institute (IRYCIS), Hospital Universitario Ramón y Cajal, 28034 Madrid, Spain; eider.pascual@salud.madrid.org; 2Medical Oncology Department, Ramón y Cajal Health Research Institute (IRYCIS), Hospital Universitario Ramón y Cajal, 28034 Madrid, Spain; javier.molinace@gmail.com (J.M.-C.); talonso@salud.madrid.org (T.A.-G.)

**Keywords:** immunotherapy, pembrolizumab, adrenocortical carcinoma, anti-PD-L1

## Abstract

Adrenocortical carcinoma (ACC) is a rare endocrine malignancy with limited treatment options in the advanced stages. Immunotherapy offers hope for altering the orthodox management of cancer, and its role in advanced ACC has been investigated in different studies. With the aim clarifying the role of immunotherapy in ACC we performed a comprehensive review about this topic focusing on the predictors of response, efficacy, safety, and the mechanisms of resistance. Five clinical trials with four immune checkpoint inhibitors (pembrolizumab, avelumab, nivolumab, and ipilimumab) have investigated the role of immunotherapy in advanced ACC. Despite, the different primary endpoints used in these studies, the reported rates of overall response rate and progression free survival were generally poor. Three main potential markers of response to immunotherapy in ACC have been described: Expression of PD-1 and PD-L1, microsatellite instability and tumor mutational burden. However, none of them has been validated in prospective studies. Several mechanisms of ACC immunoevasion may be responsible of immunotherapy failure, and a greater knowledge of these mechanisms might lead to the development of new strategies to overcome the immunotherapy resistance. In conclusion, although currently the role of immunotherapy is limited, the identification of immunological markers of response and the implementation of strategies to avoid immunotherapy resistance could improve the efficacy of this therapy.

## 1. Introduction

Adrenocortical carcinoma (ACC) is a rare endocrine malignancy with an annual incidence of 0.5–2 cases per million people [1,2]. More than half of ACC patients present locally advanced or metastatic disease [3]. The prognosis in advanced stages is poor, with a 5-year survival of 15% [4]. Moreover, in this situation, there are limited treatment options and evidence is quite scarce since although some prospective clinical studies have been carried out [5,6,7], most recommendations for ACC treatment are derived from retrospective studies. Mitotane is the only approved and consensually recommendable drug for treatment of advanced ACC [4]. Metastasectomy may be benefiting with a proper patient selection and when surgery is performed by high-volume surgeons practicing at high-volume centers [8]. Currently, systemic chemotherapy—mostly based on combination with etoposide and doxorubicin plus mitotane (EDP-M scheme)—is the most validated treatment option in advanced ACC with unfavorable prognostic parameters [4]. However, it has suboptimal efficacy and short-lived duration of disease control [9]. Radiotherapy is mostly palliative to treat selected sites of symptomatic or high-risk metastases [10]. Thus, the treatment of patients with advanced ACC refractory to “standard” therapies remains challenging. Obviously, in this setting patients should be discussed in a multidisciplinary expert team meeting with experience in care for patients with this rare disease. Apart from this, the enrolment in clinical trials based on an individual basis should be considered [4]. In this way, the collection of biological material is important with the aim of defining potential biomarkers of treatment response in the era of personalized medicine. The specific molecular alteration profiles of ACC may represent targetable events by the use of already developed or newly designed drugs enabling a better and more efficacious management of the ACC patient [11]. Molecular studies have nominated several genes as potential drivers involved in sporadic ACC tumorigenesis, including insulin-like growth factor 2 (IGF2) [12,13], β-catenin (CTNNB1) [14], and TP53 [15], among others. However, their role as a predictors of treatments response has been poorly investigated [16,17,18,19].

Regarding clinical trials investigating experimental therapies, with second-line cytotoxic regimens [9,20,21,22] the response rates are lower than 10% and median progression-free survival (PFS) is below 4 months. Neither mTOR targeting drugs nor tyrosine kinase inhibitors (TKI) are effective to avoid the early disease progression [23,24]. Furthermore, although targeting IGF2/IGF receptor signaling seemed a promising approach based on pathophysiology, the large phase III GALACCTIC trial with linsitinib has not demonstrated any improvement in progression-free or overall survival [25]. Nevertheless, a recent preclinical study suggests that the addition of mTOR inhibitors to linsitinib may increase the antiproliferative effects of linsitinib used in monotherapy [26], although it has not been clinically demonstrated. On the other hand, immunotherapy is the latest revolution in cancer therapy. However, data about the efficacy of this therapy in ACC are limited as only five clinical trials with four immune checkpoint inhibitors for the treatment of advanced ACC have been carried out. Moreover, the results regarding its efficacy are heterogeneous [27,28,29,30,31]. Nevertheless, the identification of molecular or immunological predictive factors of response may improve the antitumor immune response with these therapies [32,33,34,35]. On the other hand, mechanisms of immune resistance could be responsible for the initial disappointing results, so different strategies to overcome resistance should be considered [36,37].

Here we provide a summary of current immunotherapy ACC treatment and provides a comprehensive overview of this new therapeutic approach, including the main potential predictors of response to immunotherapy, the proved efficacy in the different clinical trials with immune checkpoint inhibitors in ACC, potential side effects, and the known mechanisms of immunotherapy resistance and potential strategies to overcome it.

## 2. Molecular Background of Adrenocortical Carcinoma

The molecular mechanisms underlying ACC onset and progression remain to be fully elucidated. Two major studies of the molecular basis of ACC—Assie et al. [12] and Zheng et al. [13]—demonstrated that loss of heterozygosity of the IGF2 locus is a common event in ACC leading to upregulation of IGF2/IGF1R signaling. Moreover, ACC shows recurrent somatic alterations facilitating rapid cell cycling, telomere maintenance, and constitutive Wnt/β-catenin and protein kinase A (PKA) signaling, in addition to those involved in chromatin remodeling, transcription, and translation [12,13], and exhibits frequent copy number alterations [13,38] (Figure 1 and Table 1).

The genomics of the adrenocortical tumors can be useful for differential diagnosis to discriminate between benign and malignant forms [11,39]. It has been shown that ACC and adrenocortical adenoma (ACA) show a differential gene expression profile, and genes involved in processes such as cell cycle or immunity are deregulated in ACC compared with ACA [15,40,41]. Among them, IGF2 is the most up-regulated gene in the malignant forms and previous studies have confirmed an overexpression of IGF2 in 90% of ACCs [12,13,15,41]. However, Heaton et al. [42] demonstrated that IGF2 overexpression probably requires additional pathways (e.g., Wnt/β-catenin signaling) for adrenocortical tumorigenesis. Gene expression profiling by transcriptome analysis identified somatic inactivating mutations of the tumor suppressor gene TP53 and activating mutations of the proto-oncogene β-catenin (CTNNB1) as frequent mutations in ACC, which seemed to be mutually exclusive and were observed only in the poor-outcome ACC group [43]. On the other hand, unsupervised clustering analysis identified two groups of malignant tumors with very different outcome based on the combined expression of PINK1 with DLG7 or BUB1B, which was the best predictor of disease-free and overall survival, respectively [44].

In terms of DNA methylation, previous studies have demonstrated that hypo and hyper-methylation alter gene expression [45,46]. Genomic studies have shown that ACCs are globally hypomethylated compared with ACAs, mainly in intergenic regions [47,48]. On the other hand, it has also been observed hypermethylated CpG islands in the promoter regions in ACC, with a possible downregulation of tumor suppressor genes [48,49]. The methylation levels of CpG islands correlates with some prognostic features and, in particular, a hypermethylated profile is associated with a poorer prognosis of ACC [12,13,49,50]. In this scenario, an altered DNA methylation status of the IGF2 locus has been associated with ACC tumorigenesis [51].

The microRNA (miRNAs) expression profile has also been shown to discriminate ACC from ACA. A deregulated expression of miRNAs has been demonstrated to alter gene expression, thus providing new tools for cancer diagnosis and prognosis [52,53,54]. Several miRNAs are differentially expressed in ACC compared to ACA, highlighting the overexpression of miR-483-5p and miR-483-3p and the concomitant down-regulation of miR-195 [55,56,57,58] and the combination of different altered miRNAs has been correlated with malignancy [55,56,59,60]. These miRNAs can thus be used to distinguish between benign and malignant adrenocortical tumors and are promising biomarkers with prognostic value in ACC patients [54,58,61].

Chromosomal alterations are also often present in ACC compared to ACA [62]. Previous analysis have shown specific amplifications in the chromosomal regions containing the TERT gene (5p15.33) and the CDK4 gene (12q14), and deletions in the chromosomal regions of the ZNRF3 (22q12.1), CDKN2A (9p21.3), and RB1 (13q14) genes [12,13,38]. Furthermore, genome analyses of ACC revealed frequent occurrence of massive DNA loss and loss of heterozygosity (LOH) followed by whole-genome doubling (WGD), which is associated with tumor aggressiveness, suggesting that WGD may represent a hallmark of disease progression [13].

Studies in the mutational landscape of ACC have allowed the identification of specific driver genes [12,13,38]. Among them, the most common altered gene is ZNRF3, which encodes an E3 ubiquitin ligase that negatively regulates the Wnt/beta-catenin pathway [12,13,38]. Other recurrently mutated genes are TP53, which is related to the cell cycle regulation, the tumor suppressor genes CDKN2A and RB1, oncogenes MDM2 and CDK4, and genes involved in chromatin remodeling (MEN1, DAXX, and ATRX) and chromatin maintenance (TERT and TERF2) [12,13,38]. Moreover, somatic mutations in genes involved in PKA activation, such as the PKA regulatory subunit PRKAR1A, have also been identified in ACC [13]. Proliferation and differentiation of the adrenocortical glucocorticoid-producing zona fasciculata is reliant on ACTH-dependent PKA signaling [63]. Additionally, cortisol-producing adenomas are characterized by abnormally high levels of PKA activation [53]. Furthermore, it was demonstrated that ACTH-dependent proliferation during zona fasciculata regeneration also relies on Wnt/β-catenin signaling [64]. This, associated with the identification of recurrent mutations leading to constitutive activation of both pathways in ACC [12,13], suggests that components of the ACTH signaling pathway may be implicated in the adrenocortical tumorigenesis. However, several genes involved in steroidogenesis are downregulated in ACC, when compared to ACA [65].

On the basis of these molecular features, it is possible to stratify ACC patients in three prognostic subgroups with different expected outcomes [11,13]. Therefore, the genomic profile allows a molecular classification of ACC and can be used to improve the diagnosis, prognosis, and management of patients with ACC, but also for the development of novel pharmacological strategies.

## 3. Markers of Response to Immunotherapy in ACC

The understanding of the molecular and immunological events underlying the pathogenesis of ACC has improved in recent years but is not yet satisfactory. Different molecular markers have been identified as potential markers of diagnosis, prognosis, and therapeutic response in ACC as we have described above (Figure 1).

Regarding immunological markers of response to immunotherapy, currently, understanding how the immune system can modulate tumor progression or effective responses against cancer is unfolding [66]. The main markers of treatment response that have been investigated in ACC are expression of like programmed death-1 (PD-1) and its ligand PD-L1, microsatellite instability (MSI) and tumor mutational burden (TMB) [67] (Table 1). Moreover, although ACC is one of the tumor types with low degree of T cell infiltration when using PD-1 mRNA expression as a marker [27], an inverse relationship of steroid hormone secretion and immune infiltration has been found [28], so treatment with glucocorticoid inhibitors drugs might increase the response to immunotherapy. Other data demonstrated high expression of the surviving protein in a series of 29 ACC. The study concluded that it could play an important role in the anti-apoptotic mechanisms in ACC and might be a new target for immunotherapy [68]. Dysfunction of TP53 due to mutations may also contribute immunologically to tumor progression and tumorogenesis, so the combination of immunotherapy and drugs targeting Wnt/beta-catenin and TP53 pathways offers promising results [37].

The Immune checkpoints, like programmed death-1 (PD-1) and its ligand PD-L1, are main regulators of T cell responses, and the use of monoclonal antibodies to block the PD-1/PD-L1 axis have shown promising results in different tumors [69]. Blocking the interaction of PD-1 with PD-L1 restored the ability of T cells to proliferate, secrete cytokines, kill infected cells [70] and may increase the immune response against tumors. Several studies have reported that the levels of PD-L1 expression both tumor cell and tumor infiltrating immune cells is a potential predictor of response to immunomodulatory agents [69,71]. The assessment of the tumor microenvironmental might help to identify those tumors more vulnerable to immunotherapy, as tumors with PD-1 expression and presence of tumor-infiltrating lymphocytes are most likely to experience response to anti-PD1/L1 blockage [72]. Nevertheless, the clinical benefit of anti-PD1/L1 was also observed in negative PD-1 tumors [73], so it is possible that other features of the immune microenvironment play a role in the PD-1/PD-L1 axis. The first study that has evaluated the prevalence and prognostic significance of PD-L1 in ACC, in a series of 28 patients with ACC, found that 10.7% of the patients were considered PD-L1 positive on tumor cell membrane and 70.4% of tumor-infiltrating lymphocytes, but no relationship to survival was observed [74]. In avelumab clinical trial [27], the proportion of PD-L1 tumoral positivity was even higher (29%). In this line of investigation, promising results have been found in a recent study of 146 ACCs, in which high PD-L1 mRNA expression was associated with biological signs of the cytotoxic local immune response, that could represent a promising strategy in “PDL1-high” ACCs, supporting the clinical trials with PD-L1-inhibitors [32]. However, in the two clinical trials with pembrolizumab [29,30] and in the avelumab study [27], no correlation was found between response to immunotherapy and PD-1 status and tumor infiltrating lymphocytes. In addition to this, in the nivolumab plus ipilimumab study [31], no information about PD-1 and PD-L1 was given; and in the nivolumab clinical trial [28], although IHC staining was positive for PD-L1 and PD-1 in 6 out of 10 patients, their association with immunotherapy response was not analyzed. This finding emphasizes the Herbst theory [73], of the role of other tumor microenvironmental factors in the PD-1/PD-L1 axis. One important factor to consider is the immunosuppressive effect of glucocorticoids in those hormone secreting-ACCs [75]. In fact, for this reason, the combination of mitotane and immunotherapy might be more potent than immunotherapy alone [9,76].

MSI is another potential marker of immunotherapy response as the damage to the mismatch repair (MRR) process leads to additive mutations along the genome, causing a “hypermutator” phenotype that presents a greater response to immunological treatments [36]. In fact, the Food and Drug Administration (FDA) approved pembrolizumab on May 23, 2017, for the treatment of patients with unresectable or metastatic, microsatellite instability-high (MSI-H), or mismatch repair deficient (dMMR) solid tumors that have progressed following prior treatment and who have no satisfactory alternative treatment options [35]. With regard to the relation between MSI and ACC, it is reported that up to 3% of all ACCs are related with Lynch syndrome [33] and 4.4% of ACCs have MSI [77]. For this reason, all ACCs should be screened for MSI and immunohistochemistry for MMR proteins and after that, germline genetic test should be performed in those patients with the absence of MMR proteins [78]. However, despite the proved efficacy of pembrolizumab in other MSI types of tumors such as melanoma, lung, kidney, and urothelial cancers, the results in ACC are limited and even controversial [9,79]. A recent study [9], with only 6 patients included, found no relation between MSI and pembrolizumab response, suggesting that mismatch protein testing may be more reliable than MSI to predict response to immunotherapy. On the other hand, a two-cases report found that the patient harboring a MSH2 mutation experienced a long-term complete response after pembrolizumab, while the other patient with high absence of mismatch repair deficiency did not have any response [79]. Nevertheless, MSI is usually no analyzed in immunotherapy clinical trials [27,28,31], as only the two studies with pembrolizumab [29,30] evaluated MSI status. Furthermore, in one of them [29] only 1 of 14 patients presented MSI, and in the other one [30] 6/39; and this last one defended that in those patients with advanced ACCs that are microsatellite stable, pembrolizumab provided similar antitumor activity than in MSI ACCs.

TMB, usually expressed as mutations per megabase with >10 being considered high, is a marker for the potential of a tumor to express and present mutant peptides in the context of human leukocyte antigen type I molecules [80]. For reactivation of T cells, mutant peptides need to be presented, so TMB could be a potential marker of response to immunotherapy. A recent study [34], found that ACC presents a high mutation rate, with a mean somatic mutation rate in the coding region of metastatic ACC of 10.17 mutations per megabase, and that ACC metastatic tumors had 2.8-fold higher median mutation rate compared to primary ACC. A recent non-randomized clinical trial [81] has explored the association of high tissue TMB with outcomes in patients with solid tumors treated with pembrolizumab. In this study, 102 of the 790 patients (13%) evaluated for TMB presented a high TMB status. Those patients with high TMB presented a higher proportion of objective response (complete or partial response) than patients with non-high TMB (29%, 95%CI = 21–39% vs. 6%, 95%CI = 5–8%). However, no patients with ACC have been included in this study. Vatrano et al. [82] performed a targeted next-generation sequencing and copy number variation analyses for 18 most frequently altered genes in ACC in 62 patients with ACC. Among other findings, they described a higher TMB in conventional and myxoid variants as compared to the oncocytic ACC. Nevertheless, to date no clinical study has focused on the implications of TMB and response to immunotherapy in ACC, and only in one of the five performed immunotherapy clinical trials [30], TMB was evaluated, and no significant relationship was observed between TMB and pembrolizumab response.

**Table 1 biomedicines-09-00304-t001:** Summary of prognostic molecular markers and immunological markers of response to immunotherapy in adrenocortical carcinoma.

**Molecular Markers of Prognosis**		
**Molecular Marker**	**Most Common Alteration/s**	**Clinical Implications**
IGF2	IGF2 overexpressed in 90% of ACCs	Targeting IGF2 system as a potential therapeutic approach [15].
[12,13,15,41]		Differential diagnosis of ACC and ACA [41].
DNA methylation	Hypomethylated intergenic regions and hypermethylated CpG islands	Hypermethylated profile is associated with a poorer prognosis of ACC [49,50].
[12,13,47,48,49,50]		Differential diagnosis of ACC and ACA (ACC are globally hypomethylated) [47,48].
microRNA	miR-483-5p and miR-483-3p overexpressed and miR-195 downregulated	Downregulation of miR-195 and upregulation of miR-483-5p are associated with poorer disease-specific survival [55,58].
[55,56,57,58]		Differential diagnosis of ACC and ACA (upregulation of miR-483-5p is a marker of ACC) [56,58].
Chromosomal alterations	Amplification in chromosomal regions of TERT and CDK4 genes, and deletions in ZNRF3, CDKN2A and RB1 genes.	Chromosomal alterations are more common in ACC than in ACA [62].
[12,13,38,62]	LOH and WGD	Copy number phenotype and WGD are hallmarks of disease progression [13].
Wnt/b-catenin pathway [12,13,38]	Abnormal cytoplasmic and nuclear accumulation of beta-catenin and somatic activating mutations of CTNNB1 and ENC1 upregulation	Activating mutations of CTNNB1 are typical of aggressive ACC [43,83].
PKA pathway	Somatic mutations in PKA regulatory subunit PRKAR1A	PRKAR1A gene mutations are typical of ACA [84].
[13,63]		Somatic activating mutations in the PKA catalytic subunit alpha gene (PRKACA) are observed in cortisol-secreting ACA [85].
**Immunological Markers OD Response to Immunotherapy**		
**Immunological Marker**	**Most Common Alteration/s**	**Clinical Implications**
PD-1/PD-L1 [27,28,29,30,31,69,71,74]	10.7% of ACCs are PD-L1 positive on tumor cell membrane, 70.4% on tumor-infiltrating lymphocytes.	Levels of PD-L1 expression as a potential predictor of response to immunomodulatory agents [69,71].
MSI [33,36,77]	3% of all ACC are associated to Lynch syndrome and 4.4% have MSI.	MSI may causing a “hypermutator” phenotype that presents a greater response to immunological treatments [36]
TMB	High TMB status in ACC.	ACC metastatic tumors had 2.8-fold higher median mutation rate compared to primary ACC [34].
[30,34,82]		Higher TMB in conventional and myxoid variants than in oncocytic ACC [82].

ACC: Adrenocortical adenoma; ACC: Adrenocortical carcinoma; CDK4: Cyclin Dependent Kinase 4; CDKN2A: Cyclin dependent kinase inhibitor 2A; ENC1: Ectodermal-Neural Cortex 1; IGF2: Insulin-like growth factor 2; LOH: Loss of heterogeneity; MSI: microsatellite instability; PD-1: Programmed Death 1; PD-L1: Programmed Death-ligand 1; PKA: Protein kinase A; PRKAR1A: Protein kinase cAMP-dependent type I regulatory subunit alpha; RB1: Retinoblastoma 1; TMB: Tumoral mutation burden; TERT: Telomerase reverse transcriptase; WGD: Whole-genome doubling.

## 4. Immunotherapy in Adrenocortical Carcinoma: Efficacy

Immunomodulation has been analyzed in patients harboring metastatic ACC due to the lack of effective and safe new drugs in this setting and the potential development of adrenalitis as an adverse event in patients receiving treatment with immune checkpoint inhibitors (ICI). Indeed, preclinical findings suggest a potential benefit from this therapeutic strategy in advanced ACC [86] (Figure 2).

### 4.1. Programmed Cell Death Protein 1 (PD-1) and Programmed Cell Death Protein Ligand (PD-L1) Axis and PD-1/PD-L1 Blockade

#### 4.1.1. Pembrolizumab 

Raj, et al. [30] developed an investigator-initiated study including patients with advanced ACC receiving pembrolizumab, with mitotane discontinuation. Molecular analysis included PD-L1 expression by IHC (Positive: ≥1% in tumor or tumor stromal interface), a tumor-infiltrating lymphocyte (TIL) score, MMR status by immunohistochemistry (IHC) and Next Generation Sequencing (NGS) with MSI-H status and TMB > 10 mutations/megabase. Once 3 patients in the stage I of the study showed an objective response, recruitment continued to 39 patients. After a median follow up of 17.8 months (range, 5.4–34.7), the ORR was 23% (9/39) with all of them being partial responses with a time to response of 4.1 months (range 1.7–10.5). Disease control rate (DCR) was 52% (95% CI, 33% to 69%). Translational research showed a PD-L1 expression in 7 of 34 patients analyzed with no significant difference in response according to PD-L1 status. Additionally, 6 of 38 patients had MSI-H/MMR-D tumors with a partial response in 2 patients and progressive disease in 2 patients. No relationship was found between somatic alterations analyzed by NGS, TMB, or TIL score and response to pembrolizumab (Table 2).

Another phase II trial was performed [29] at the MD Anderson Cancer Center including 16 patients with advanced ACC (7 patients had cortisol-producing disease) (NCT02721732). No concomitant oncology therapy was allowed. The primary endpoint, the non-progression rate at 27 weeks, was 36% (95% CI 13% to 65%). Molecular analysis showed no expression of PD-L1 in any tumor sample analyzed. In addition, 13 tumor specimens were microsatellite-stable and 8 tumor samples showed high TILs. Therefore, no association between potential predictive biomarkers and response to pembrolizumab could be determined (Table 2).

The combination of ICI, such as pembrolizumab, with cytotoxic chemotherapy has been evaluated in a small sample of 12 patients [87]. Four patients received concomitant cyclophosphamide and gemcitabine plus docetaxel and 5 patients received concomitant steroidogenesis inhibitors, such as metyrapone and ketoconazole. Median PFS was 1.4 months and median OS was 5.3 months. Though no impressive result was identified for the whole population, the combination therapy was found safe. The combination of pembrolizumab with mitotane was analyzed in a retrospective series including 6 patients with advanced ACC that had previously received treatment with chemotherapy (3 patients) and mitotane (6 patients) [88]. Two patients harbored an MSH2 mutation within a Lynch syndrome. The results were promising with clinical benefit in all patients treated with 2 partial responses and 4 stable diseases according to RECIST criteria. The preclinical rational for a synergistic activity between ICI and mitotane is required, as it has been suggested by tumor responses achieved in different clinical studies. Lenvatinib is another partner for pembrolizumab combination in a retrospective series with heavily pretreated metastatic ACC patients [89]. The potential immunomodulation induced by lenvatinib has been suggested as a potential mechanism of synergy between this drug and ICI. Two patients achieved a partial response and 1 stable disease. The median PFS was 5.5 months (95% CI 1.8–not reached) (Table 2). Previously mentioned trials arouse the question of different tumor behavior to immunotherapy treatment according to the presence of hormone producing tumors or non-functioning tumors. However, no preclinical analysis has justified this hypothesis yet.

#### 4.1.2. Nivolumab 

Ten patients with advanced ACC (4 hormone-producing tumors), including 2 patients previously untreated, received treatment with nivolumab [28]. The ORR according to RECIST 1.1 criteria showed one unconfirmed partial response (due to treatment withdrawal related to adverse events) and 2 stable diseases. After a median follow up of 4.5 months, the 6-month OS rate was 56% (95% CI 8% to 88%). The study included in IHC expression analysis from PD-L1, PD-1, CD8, and CD4 showing an important heterogeneity. The molecular analysis and flow cytometry of immune cells was performed in only 5 patients and no conclusion could be drawn (Table 2).

#### 4.1.3. Avelumab 

Le Tourneau C, et al. [27] published the greatest trial analyzing the role of ICI in advanced ACC with avelumab, a PD-L1 inhibitor. The study included 50 previously treated patients with a median of 2 lines (range, 1 to 6) and 37 patients had received ≥ 2 prior lines. Concurrent treatment with mitotane was given in half of patients. After a median follow up of 16.5 months, the ORR was 6% (95% CI, 1.3% to 16.5%), median PFS was 2.6 months (95% CI, 1.4 to 4.0) and median OS was 10.6 months (95% CI, 7.4 to 15.0). Though no definitive conclusion can be made, better outcomes were identified in patients with a lower number of prior treatment lines and PD-L1 positive expression. In this sense, considering PD-L1 expression on tumor cells cut-off ≥5%, median PFS was 5.5 months in patients with PD-L1 positive tumors (N = 12) vs. 1.7 months in patients PD-L1 negative tumors (N = 30) (HR = 0.66; 95%CI 0.32–1.39). Mitotane was safely administered in combination with avelumab (Table 2).

### 4.2. Cytotoxic T-Lymphocyte-Associated Antigen 4 (CTLA-4) Blockade

#### Ipilimumab 

The combination of PD-1 with CTLA-4 inhibitors is based on a potential synergistic effect that has been reported in preclinical and clinical studies [90] (Table 3). There are tumor types where it is necessary to promote the activation of naive T lymphocytes and other cells sequentially involved in the activation of the immune response at the level of the lymph node and peripheral tissues, in addition to overcoming the immunosuppressive pathways in the tumor microenvironment. In this sense, if CTLA-4 initially acts at the level of T lymphocyte activation and PD-L1 does so at the tumor, the combination of both could have a synergistic effect [91]. Therefore, the combination of CTLA-4 and PD-1/PD-L1 inhibitors seeks to increase immune activation at the level of lymph nodes and in peripheral tissues, in addition to reversing the activity of exhausted T lymphocytes.

In advanced ACC, the combination of nivolumab and ipilimumab has been analyzed in a phase II multicohort trial of rare genitourinary cancers (NCT03333616) [92]). Sixteen patients with metastatic ACC were included in the study. Four patients were treated in first line and most of them (N = 9) had received one line of prior treatment. The patients included in this cohort achieved an ORR of 6% (1/14) and a Disease Control Rate (DCR) of 53.3% (8/15). The median PFS was 4.5 months (95% CI 1.8–6.6) and 12 month rate was 43% (95% CI 8–75%). Those results suggest further research is required in this tumor to better characterize the role of this ICI combination in the therapeutic algorithm. The most frequent adverse events were liver abnormalities (38%), fatigue (36%), all rashes (35%), thyroid disorders (24%), and pruritus (22%).

### 4.3. Potential Immune Related Targets under Research in ACC 

Initial results with PD-1/PD-L1 and CTLA-4 inhibitors have been the starting point for immune-based therapy research in advanced ACC monotherapy or in combination due to the safety profile (Table 3). One of the most promising strategies at this moment is the activation of memory T-cells that respond to microbiome-derived peptide antigens to direct an effector immune response against the tumor. This is currently under research with the cancer peptide vaccine EO2401 whose tumor antigens are expressed in glioblastoma and adrenal tumors (NCT04187404). Other molecules involved in immune response regulation are chemokines and their receptors. CXCL12-CXCR4 axis has been related to poor oncological outcomes and resistance mechanisms of different cancer therapies, such as PD-1/PD-L1 immunotherapies [93]. To date, there is only one CXCR4 antagonist (plerixafor^®^) approved in the treatment of hematologic malignancies. In this sense, CXCR4 and CXCR7 have been analyzed as potential targets in advances ACC, but further clinical trials are required [94]. Furthermore, ICI are safely administered in combination with other targets that may help to increase the activity of PD-1/PD-L1 monotherapy. Drugs to combine that are currently under research act at different pathways, such as VEGFR-driven TKI (cabozantinib) in the CABATEN trial (NCT04400474) or PI3K/Akt pathway inhibition (eganelisib or IPI-549 as a selective inhibitor of PI3Kγ) in a phase II trial (NCT02637531).

**Table 2 biomedicines-09-00304-t002:** Summary of studies that have investigated immunotherapy in patients with adrenocortical carcinoma.

Drug	Study Design	Population	Number of Patients	PD-L1 Status (IHC)	Primary Endpoint	Other Main Endpoints
Pembrolizumab 200 mg every 3 weeeks during 24 months (35 cycles)	Phase II-single arm [30]	Prior systemic therapy: 28 patients (31% with ≥1 prior line)	39	7/34	ORR RECIST 1.1 = 23%	DoR = NR PFS = 2.1 months OS = 24.9 months
Pembrolizumab 200 mg every 3 weeeks during 24 months (35 cycles)	Phase II-single arm [29]	Prior systemic therapy: median number of prior lines = 2 (1–5)	16	0/14	Non-progression rate at 27 weeks = 36%	ORR = 14%
Pembrolizumab 200 mg every 3 weeeks + Mitotane	Retrospective [88]	Prior 1 line of systemic therapy	6	NA	NA	Two patients PR and four SD
Pembrolizumab 200 mg every 3 weeeks + Lenvatinib	Retrospective [89]	Prior systemic therapy	8	NA	ORR = 25%	PFS = 5.5 months
Nivolumab 240 mg every 2 weeks	Phase II-single arm [28]	Prior 0—≥1 cisplatin-based chemotherapy	10	6/10	ORR RECIST 1.1 = 11%	PFS = 1.8 months
Nivolumab 3 mg/kg plus Ipilimumab 1mg/kg	Phase II—multicohort [31]	Prior 0—≥1 cisplatin-based chemotherapy	16	NA	ORR RECIST 1.1 = 6%	PFS = 4.5 months
Avelumab 10 mg/kg every 2 weeks	Phase Ib expansion cohort [27]	Prior cisplatin-based chemotherapy. Concomitant mitotane allowed.	50	12/42	ORR RECIST 1.1 = 6%	PFS = 2.6 months OS = 10.6 months

ORR: Overall Response Rate; DoR: Duration of Response; PFS: Progression Free Survival; OS: Overall Survival; NR: Not reached; NA: Not available.

**Table 3 biomedicines-09-00304-t003:** Ongoing clinical trials with immunotherapy agents in advanced Adrenocortical carcinoma (ACC).

Study Design	NCT Identifier	Treatment	Estimated N	Primary Endpoint
DART trial Phase 2 multicohort	NCT02834013	Nivolumab + Ipilimumab	818 (all cohorts)	ORR RECIST 1.1 in subsets
Phase 2 multicohort	NCT02721732	Pembrolizumab	225 (all cohorts)	Non-progression rate Incidence adverse events
Phase I/II	NCT04187404	EO2401 + Nivolumab	60	Incidence adverse events
Phase I/Ib first-in-human multicohort	NCT02637531	Nivolumab + Eganelisib	219 (all cohorts)	Dose limiting toxicities Adverse Events
Phase II multicohort	NCT04400474	Cabozantinib + Atezolizumab	144 (all cohorts)	ORR RECIST 1.1

ORR: Overall Response Rate.

## 5. Safety of Immunotherapy

Blocking of PD-1/PD-L1 immune checkpoint leads to the development of new toxicities by reactivation of the immune system [95,96]. Immunotherapy is usually well tolerated but some patients could develop Immune Related Adverse Events (IRAE). The most common observed side effects affected skin, endocrine and gastrointestinal system and are generally mild [30]. Nevertheless, some variables have been associated with a higher risk of develop severe side effects, such as patient sex, a history of autoimmune disease, previous treatment with anti-CTLA-4 inhibitors, kidney failure, treatment with glucocorticoid previous to the initiation of immunotherapy and the use of combined therapy with anti-CTLA-4 and anti-PD-1 [97,98]. IRAE occur quite early, mostly within weeks to 3 months after initiation of immune checkpoint blockers. However, this IRAEs can occur at any time, from the outset of treatment, during treatment, or after treatment has been discontinued. Skin irAEs usually are the first ones to develop, followed by gastrointestinal toxicities. Hepatitis and hypophysitis may develop later in time [99]. Physician education and patient awareness of IRAEs is key to reducing the severity of these events. Timely intervention with corticosteroids, in serious IRAEs, is crucial to limit the severity of these events. A multidisciplinary approach, depending on localization of IRAE, is a mainstone in its management [100].

In patients treated with anti-PD-1, gastrointestinal IRAEs occur in <20% [101], skin reactions in 30–40% and immune-related hepatitis in 5% [102]. Regarding endocrine irAEs, thyroid dysfunctions are the most frequently observed (in 4% of the patients) [103]. Other less common endocrine irAEs are hypophysitis in 1% and primary adrenal insufficiency, type 1 diabetes mellitus, hypercalcemia and hypoparathyroidism in less than 1% [104]. CTLA-4 inhibitors seems to produce IRAEs more frequently compared with PD-1/PD-L1 inhibitors [105]. In this way, gastrointestinal and skin irAEs occur in 44% and 50%, respectively [102,106]. Furthermore, endocrine irAEs are more common than with anti-PD-1/PD-L1 inhibitors, observing hypophysitis in 13% and hypo-/hyperthyroidism in 6%. Other endocrine IRAEs are also rarely reported.

## 6. Mechanisms of Resistance to Immunotherapy in Adrenocortical Carcinoma

In general, lymphocyte activation is an extremely complex process that requires perfect synchronization of the immune system. First, the presentation of the antigen by the antigen-presenting cell towards the T-lymphocyte receptor. This lymphocyte synapse is not enough, co-stimulation between the CD28 of the T-lymphocyte and the B7/B7.1 of the antigen-presenting cell is required to complete the T cell activation. At this time, the activated T lymphocyte can exert its cytotoxic function.

Furthermore, once activated, the lymphocyte has to locate the target antigen by crossing the tumor microenvironment. But the immune system has generated mechanisms to modulate the effector response of cytotoxic T lymphocytes, to avoid deleterious autoimmunity effects. This means that at any point of this migration, the cytotoxic T lymphocyte can be inactivated by the cells of the tumor microenvironment. Once it reaches the target cell, it must be able to exert its cytotoxic function on, again avoiding its inactivation. These general steps must be fulfilled in the activation and cytotoxicity process of lymphocytes stimulated by immunotherapy. Any alteration in one of the steps will lead to the failure of its cytotoxic activity [100].

More particularly, mechanisms of resistance to immunotherapy have been described in adrenal cancer. First, PD-L1 expression in ACC is low, being around 10% of ACC tumors that overexpressing PD-L1. It is known that correlation between tumor PD-L1 expression and response to PD-1 therapy has been provided for various cancer types [37,74]

Second, molecular alterations that lead to an altered production of CD8 + infiltrate thus impairing the local antitumor immune response were described in ACC. Alterations in WNT-β—catenin pathway, consisting in upregulation of β-catenin, have been associated with a reduced recruitment of the specific lineage basic leucine zipper transcriptional factor ATF-like 3 lineage (BATF3) of dendritic cells. This lineage of dendritic cells are associated to the production of chemokines, such as the CXCL9 and the CXCL10, directly related with high number of infiltrating T cells. Moreover, TP-53 mutations lack production of key chemokines required for the recruitment of T cells and natural killer, which results in T cytotoxic cell exclusion from the tumor infiltration [36,107]. Upregulation of β-catenin and TP-53 inactivating mutations lead a production reduced of chemokines necessaries to cytotoxic activation and migration. Both alterations represent an oncology target challenging nowadays.

Lastly, glucocorticoids play a key role in ACC resistance to immunotherapy. One the one hand, clinical behavior in patients with cortisol producing ACCs are characterized by lower levels of circulating lymphocytes and suppression of T cytotoxic cell [37,108]. On the other hand, glucocorticoid supplementation in patients with adrenal deficiency treated with mitotane or after adrenal surgery have the potential to impair immunotherapy efficacy in ACC patients by immunosuppressive activity [88,109].

## 7. Conclusions

Nowadays, only four immune checkpoint inhibitors have been tested for the treatment of ACC in clinical trials. Results regarding its efficacy are heterogeneous, but usually with low rates of overall response and progression free survival. Thus, currently the role of immunotherapy in ACC is limited. However, the identification of immunological markers of immunotherapy response and the implementation of strategies to avoid immunotherapy resistance could lead to a greater efficacy of this treatment, making that immunotherapy could be a new promising therapeutic option in properly selected patients with advanced ACC.

## Figures and Tables

**Figure 1 biomedicines-09-00304-f001:**
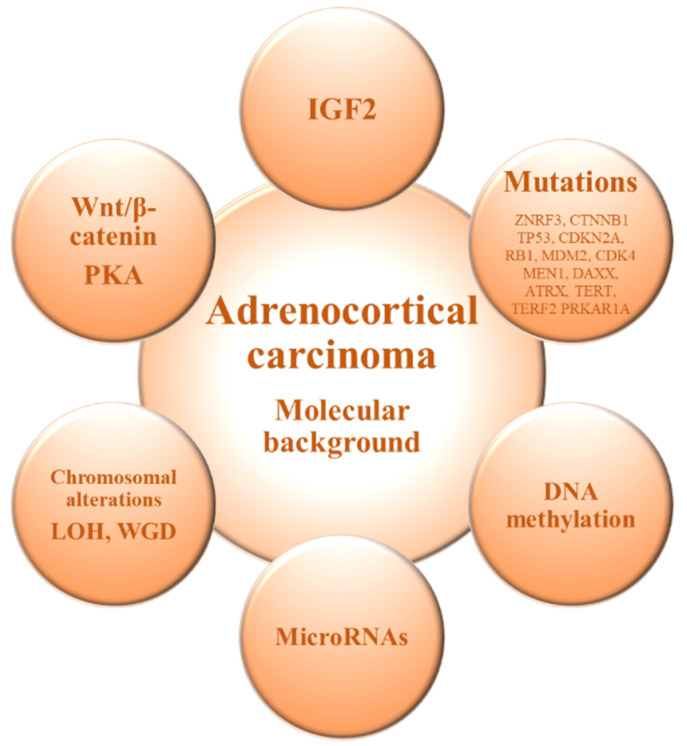
Molecular alterations in adrenocortical carcinoma. IGF2: Insulin-like growth factor 2; LOH: Loss of heterozygosity; PKA: Protein kinase A; WGD: Whole-genome doubling.

**Figure 2 biomedicines-09-00304-f002:**
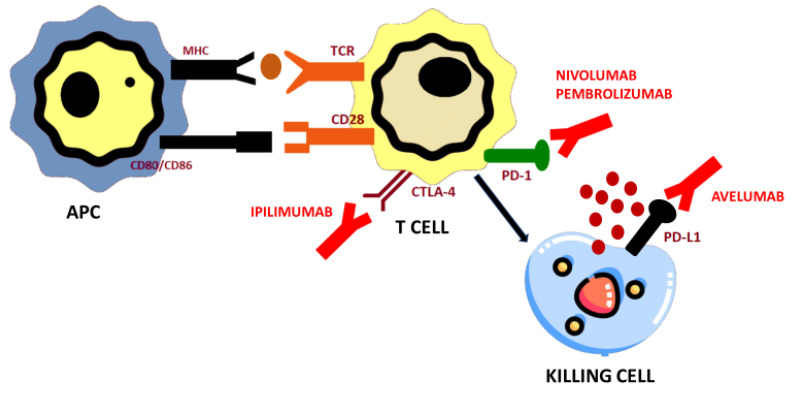
Mechanism of action of programmed death-1 (PD-1)/PD-L1 and CTLA-4 immuno checkpoint inhibitors used for adrenocortical carcinoma treatment. Immunotherapy targeting CTLA-4 (ipilimumab), PD-1 (nivolumab, pembrolizumab), and PD-L1 (avelumab) block immune checkpoints (CTLA-4, PD-1, and PD-L1, respectively) and restore antitumor immune response, resulting in tumor cell death via release of cytolytic molecules (e.g., granzyme B, TNF-a, INF-g). APC: professional antigen presenting; TCR: MHC-T cell receptor.

## Data Availability

Not applicable.

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
