# Peer review of "Immunotherapy in Adrenocortical Carcinoma: Predictors of Response, Efficacy, Safety, and Mechanisms of Resistance"

_biomedicines, 2021, doi:10.3390/biomedicines9030304_

Round 1

Reviewer 1 Report

The review entitled “Immunotherapy in Adrenocortical Carcinoma: predictors of response, efficacy, and mechanisms of resistance” by Araujo-Castro and coll deals with an interesting topic related to the medical treatment of a very rare and aggressive cancer, with a dismal prognosis, such as AdrenoCortical Carcinoma.

The same topic, however, is widely presented and discussed in a series of very recent (2019-2020) reviews:

J Endocrinol Invest. 2020 Nov;43(11):1531-1542; Eur Urol Focus. 2020 Jan 15;6(1):14-16; Ann Endocrinol (Paris). 2020 Dec 3;S0003-4266(20)31307-X; Future Oncol. 2020 Dec;16(36):3017-3020; J Oncol. 2019 Apr 1;2019:6072863).  

The most recent review is published on Biomedicines in January 2021: Biomedicines. 2021 Jan 20;9(2):E98. doi: 10.3390/biomedicines9020098

I therefore believe that, at the moment, the presente review will not add any new findings compared to what it is already known.

Furthermore, in the Introduction, the sentence “most recommendations for ACC treatment are derived from retrospective studies”   is not completely  correct as prospective clinical studies related to ACC therapy have been published, i.e. with the EDP-M approach or  with cabozantinib, to cite few.

Author Response

Dear reviewer,

I attached the answers to your comments in this word document

Kind regards

Reviewer 2 Report

This review is comprehensive and overall clear.

I would reduce the first part on molecular pathogenesis of ACC, focusing on what may be pertinent to immunotherapy.

I would tone down the last statement claiming that immunotherapy is very promising while in the same paragraph it is said that results are up to now limited. It should be better said that immunotherapy could be promising if we will be able to circumvent resistance to treatment, etc.

Author Response

(The authors gave the same response as above.)

Reviewer 3 Report

This is a very interesting and well-written review. The authors provide a comprehensive report on the immunotherapy options in adrenocortical cancer. The text is clear, the tables are well-built and are easy to follow.

The manuscript dissects an important field in adrenocortical cancer treatment. The pathogenesis of ACC is briefly presented, then the issues linked to immunotherapy in detail.

There are few spelling errors in the manuscript.

I would suggest that a Figure on the way of action of immune checkpoint inhibitors could be (PD1-PDL1, CTLA-4) could further enhance the quality of the manuscript.

I would also suggest to include a brief summary on the potential side effects of immune checkpoint inhibitors, as these even include endocrine side effects (e.g. hypophysitis, thyroiditis, adrenocortical insufficiency) that could have further relevance in patients with ACC.

Author Response

Dear reviewer,

I attached the answers to your comments in the word document

Kind regards

Round 2

Reviewer 1 Report

Thanks for the reply